# Random Scaling of Emergent Capabilities

**Rosie Zhao** [* 1 2]   **Tian Qin** [* 1]   **David Alvarez-Melis** [1 2]   **Sham Kakade** [1 2]   **Naomi Saphra** [1 2]

## Abstract

Language models famously improve under a smooth scaling law, but some specific capabilities exhibit sudden breakthroughs in performance. Advocates of "emergence" view these capabilities as unlocked at a specific scale, but others attribute breakthroughs to superficial metric thresholding effects. We propose that breakthroughs are instead driven by continuous changes in the *probability distribution* of training outcomes when performance is bimodally distributed across random seeds. we show that different random seeds can produce *either* smooth *or* emergent scaling trends in synthetic length generalization tasks, multiple choice question answering, and grammatical generalization. We reveal that sharp breakthroughs in metrics are produced by underlying continuous changes in their distribution across seeds. These distributions may become abruptly bimodal at a capacity threshold—but this threshold appears at scales well before most seeds achieve breakthrough. Our observations hold true even under continuous loss metrics, confirming that random variation must be considered when predicting a model's performance from its scale.

## 1. Introduction

On most benchmarks, language model (LM) performance is determined by a scaling law (Hestness et al., 2017; Rosenfeld et al., 2019; Kaplan et al., 2020) that varies smoothly with parameter size and overall training compute. There are, however, a number of celebrated exceptions in which performance abruptly improves on specific benchmarks (Srivastava et al., 2023). These sudden breakthroughs fuel one of the most heated debates in modern AI.

*Equal contribution [1] Harvard University. [2] Kempner Institute for the Study of Natural and Artificial Intelligence. Correspondence to: Rosie Zhao <rosiezhao@g.harvard.edu>, Tian Qin <tqin@g.haravrd.edu>, Naomi Saphra <nsaphra@fas.harvard.edu>.

*Proceedings of the 43rd International Conference on Machine Learning*, Seoul, South Korea. PMLR 306, 2026. Copyright 2026 by the author(s).

On one side, advocates of **emergence** claim that performance abruptly improves when a particular scale provides the capacity to learn specific concepts (Wei et al., 2022). On the other side, skeptics argue that these sudden improvements are a **mirage** (Schaeffer et al., 2024) driven by thresholding effects. These threshold artifacts are alleviated by more appropriate continuous metrics—though a few **breakthrough capabilities** remain stubbornly emergent. We argue that discontinuities are driven by continuous changes in the *probability* of a breakthrough at each scale. In other words, the discontinuities are real—each model firmly either *knows* or *does not know* a given concept—but breakthroughs do not always reflect a fixed threshold at which a concept is learnable. Instead, models may learn the concept at various scales, albeit with changing probability.

We posit that a breakthrough capability is distinguished not by deterministic responses to scale, but by *multimodal* random variation. In other words, independent training runs cluster in their performance metrics. This observation is undocumented because scaling laws usually plot a single training run at each scale, rarely testing multiple seeds. Although random variation may be benign when model performance is measured in-distribution (Jordan, 2024), previous work suggests that out-of-distribution performance may vary widely across training runs (Zhou et al., 2024a;b; Qin et al., 2024; Juneja et al., 2023; Li et al., 2025), even at larger scales (Madaan et al., 2024).

By connecting breakthrough scaling with random variation, we challenge the narratives of both the *emergence* and *mirage* camps. First, our results *contest the position of the mirage or "loss-to-downstream" camp*: that effects of scale are predictable and that continuous metrics will smooth out apparent breakthroughs. We discover clustered multimodal distributions of capabilities, confirming that models unpredictably learn critical discrete concepts—and these clusters are observable even using continuous loss metrics. Furthermore, we *complicate the rival narrative of the emergence camp*, which often treats specific model sizes as distinct in their capacity. While we confirm that algorithms require some minimum model capacity, that capacity may not be the enormous scale at which emergence is observed. Instead, *discontinuous* performance jumps are sampled from a *continuously* changing multimodal distribution, where the "success" mode ultimately dominates at larger scales. To

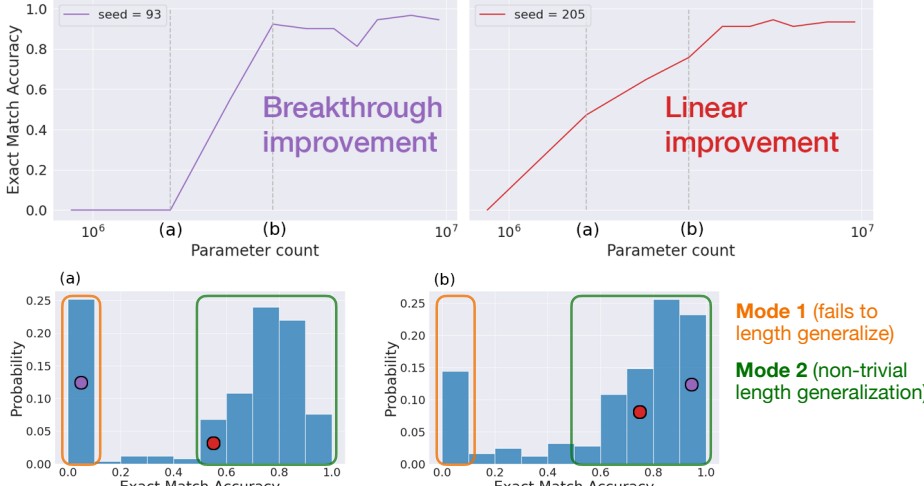

*Figure 1.* **Different random seeds produce different scaling trends.** Scaling trends can be emergent or linear for different seeds, even if all models train on the same data with the same hyperparameters. On the count task (see Appx. C), we show trends for random seeds with the highest breakthroughness (seed 93; top left) and linearity (seed 205; top right). We mark parameter counts immediately before and after seed 93's emergence, respectively, as (a) and (b). Histograms illustrate the bimodal distribution of performance across all random seeds at scales (a) and (b), marking the positions of seeds 93 and 205. Breakthroughs occur when consecutive points represent different clusters; linear trends occur when each point is sampled from the same gradually shifting cluster. We mark the two modes in orange and green, where the former consists of runs which completely failed to length generalize, and the latter consists of runs where non-trivial length generalization was achieved.

extrapolate downstream metrics to large scales, we must predict the likelihood of successful emergence as well as the performance of a successful run.

Since training numerous seeds is prohibitively expensive at large scales, we study partially reinitialized LLMs and toy models. Whereas prior work reports summary statistics across only a few training runs, we characterize the full multimodal performance distribution. We find:

- **Breakthroughs result from bimodal performance distributions.** On synthetic algorithmic tasks (Sec. 2), simple language modeling tasks (Sec. 4) and multiple choice question answering (Sec. 3) , some random seeds produce linear scale trends while others are emergent. This variation is caused by the bimodal distribution of each skill across seeds (Sec. 2.2), a property that materializes around **breakthrough thresholds** in model size. At these scales, emergence is a stochastic property (Sec. 2.3).
- **Bimodal variation persists under continuous metrics.** In the emergence debate, one main position (Schaeffer et al., 2024) is that breakthroughs are caused by measuring discontinuous metrics such as exact match accuracy rather than loss. We find continuous loss metrics can remain visibly bimodally distributed, particularly when there is a more even split of failed and successful runs. We confirm these results on both synthetic (Sec. 2.4, Sec. 4.2) and natural (Sec. 3.3) tasks.
- **When a scale curve exhibits sudden *discontinuous***

**improvement in a skill, the *probability* of learning that skill may be changing *continuously***. Treating the bimodal distribution as a mix of **failure** and **success** distributions, we illustrate that average improvements can come from changes in the probability of success *or* in the mean performance of a successful run (Sec. 2.2). Although bimodality can appear abruptly at a minimum capacity scale (Sec. 2.5), these sudden distributional changes do not necessarily align with breakthrough scales for individual model runs.

## 2. Synthetic Length Generalization Tasks

Usually, scaling trends are measured on a single model per scale or, at most, the average of a few runs. The literature suggests that emergent capabilities unlock at specific model scales (Wei et al., 2022), implying that different training runs would perform similarly at each scale. Contrary to this belief, we demonstrate that performance is sampled from a stochastic distribution which changes gradually even as individual scaling curves jump abruptly.

### 2.1. Tasks and Setup

After training models on synthetic tasks, we measure their length generalization (Graves et al., 2016; Kaiser & Sutskever, 2015; Lake & Baroni, 2018; Hupkes et al., 2020), one of many compositional skills that can lead to conceptual breakthroughs (Srivastava et al., 2023; Löwe et al., 2024;

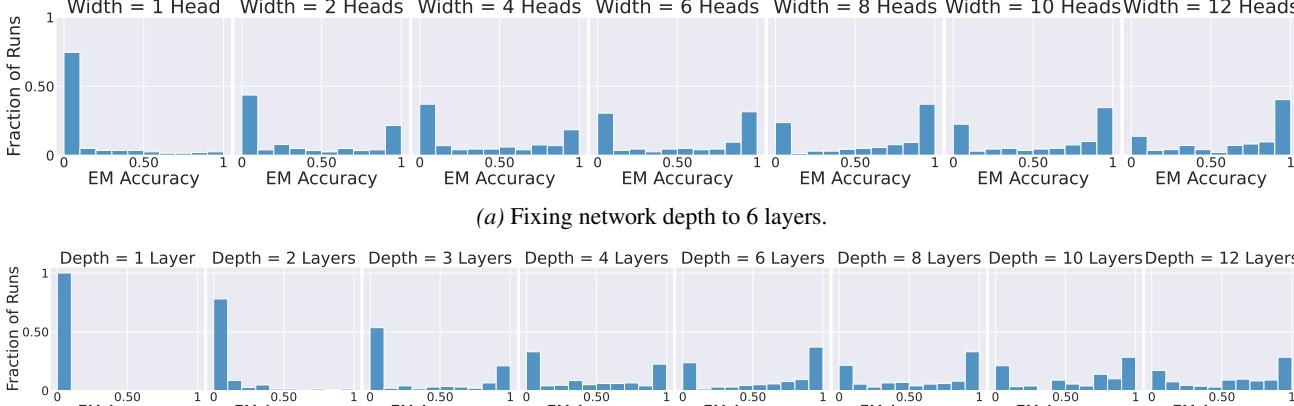

*(a)* Fixing network depth to 6 layers.

*(b)* Fixing network width to 8 heads per layer.

*Figure 2.* **Random variation in length generalization (addition task).** Histograms of exact match accuracy on length 40 sequences when independently scaling a width and b depth.

Chen et al., 2024). Experimental details are in Appx. A.

**Architecture:** In our synthetic experiments, we train decoder-only Transformer models from scratch using rotary position embeddings (RoPE) (Su et al., 2024). To observe the random performance distribution at each scale, we train our models from hundreds of seeds. We choose model sizes by separately adjusting either the width (number of 64-parameter heads per layer) or depth hyperparameter.

**Task:** We consider two algorithmic tasks previously studied in Zhou et al. (2024a): counting and addition. An analysis of the **count** task is provided in Appx. C, including discussion (Appx. C.2) of how its average accuracy responds non-monotonically to scale. In the main body of this paper, we will focus on the **addition** task. Zhou et al. (2024b) showed that Transformers can generalize 10-15 digits past training length for an addition task, if provided with index hints and allowed to generate the answer backwards. We use this modified reverse-order addition task.

**Dataset:** During training, we sample sequences i.i.d from the train set and invoke in-context learning by adding examples to the context, following prior work (Jelassi et al., 2024; Zhou et al., 2024b). The lengths of examples are sampled uniformly from 1 to the maximum training length (30 for count and 35 for addition). Length generalization is then tested at length 60 for count and 40 for addition.

### 2.2. Emergence is a Sign of Bimodal Variation

What do the scaling curves for length generalization tasks look like when we generate them in conventional ways? Following Srivastava et al. (2023), we calculate the *breakthroughness* and *linearity* of scaling curves where each run

shares a fixed initialization and shuffle seed.[1] As defined in Appx. B, breakthroughness measures emergence, whereas linearity measures a smooth response to scale. We plot the performance across scale for the seeds with the highest breakthrough and highest linearity in Fig. 1 (with other extremes shown in Appx. Fig. 11). Because breakthroughs vary across seeds, we can easily find fixed seeds that lead to *either* emergent *or* smooth scaling.

This variation is explained by the bimodality of model performance distributions when varying seed. Fig. 2 illustrates that, for a population of models independently trained on the addition task, length generalization ability clusters into high and low component modes at many parameter sizes. This clustering produces distinctly bimodal performance distributions, causing some model runs to appear as breakthroughs while others generalize poorly. When bimodally distributed runs cluster into distinct high- and low-performance components, a model might exhibit linear scaling if sampled from the same cluster as the previous scale *or* emergent scaling when switching from the low cluster to the high cluster. These differences ultimately lead to high variability in the timing and degree of emergence. Furthermore, these differences cause oscillating scale curves like those seen in low-linearity tasks (Srivastava et al., 2023, ref. Fig. 7c).

### 2.3. Sudden Jumps From Gradual Distribution Shifts

When reporting metrics from only one seed or the mean of a few seeds, the outcome is likely to be close to the *mode* performance of the underlying model population. In Fig. 3 (top left), the mode performance shows a massive spike,

---

[1]Although it is standard practice to fix the random seed when comparing LM benchmark performance across scales, the initializations produced by a single seed have no meaningful relation across different model sizes.

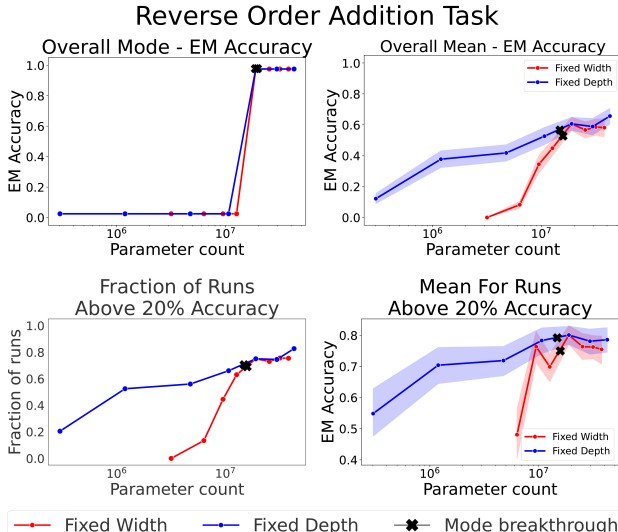

*Figure 3.* **Summary statistics for length generalization accuracy on the reverse-order addition task.** We track overall mode (top left) and overall mean (top right). While the mode exhibits a sharp increase in accuracy at a certain scale of width or depth, the mean evolves more continuously as a result of the bimodal nature of the random variation distribution. We also study successful runs, plotting the fraction of runs reaching above 20% accuracy (bottom left) and the mean of such runs (bottom right). The plots for mean include 95% confidence intervals with 1000 bootstrapped samples.

mirroring an emergent benchmark's scaling trend. However, we claim that this discontinuous improvement in mode—as well as the discontinuities seen in single-seed emergent scaling curves—is an artifact of bimodality. Underlying this *discontinuous* performance jump are *continuous* changes in other distributional statistics. In Fig. 3 (top right), the mean exhibits a smoother trend in accuracy. Mode and mean differ because the underlying distribution is bimodal, expressing a mixture of "successful" and "failing" runs.

Treating the distribution as a mixture of successes and failures, we can separately analyze the *probability* of a successful run and the *performance distribution* of successful runs. Both of these properties are changing continuously and gradually when the mode increases abruptly. If we restrict our analysis to the runs achieving at least nontrivial 20% accuracy, we see that the probability (Fig. 3 (bottom left)) and mean (bottom right) of such "successful" runs both exhibit continuous improvement, with the exception of increasing from depth 2 to 3 (discussed further in Sec. 2.5). Even at the mode breakthrough, these underlying distributional properties are only changing gradually. We conclude that *when tasks exhibit bimodal distributions across random seeds, gradual statistical improvements cause seemingly abrupt improvements with scale.*

## 2.4. Is Bimodality a Mirage?

Metrics with hard thresholds can artificially induce breakthroughs (Schaeffer et al., 2024); conversely, continuous metrics turn apparent emergence into smooth curves (Srivastava et al., 2023). We must be particularly cautious about claiming emergence when requiring outputs to exactly match a target string, as we have so far. Are our case studies artifacts of thresholding effects? Do our bimodal distributions become unimodal under continuous metrics?

To avoid thresholding artifacts, consider a continuous equivalent to the exact match metric: the maximum loss assigned to any individual token. Each token is individually computed in the correct output context, so the entire sequence represents a fixed set of continuous per-token loss functions. Because the maximum of a fixed set of Lipschitz-continuous functions is always Lipschitz-continuous, this error score is guaranteed to be continuous as long as per-token loss is continuous. For model $f$ on dataset $X$, the continuous error score based on per-token loss $L(f(x_{0...i-1}), x_i)$ is:

$$\text{error}(f, X) = \frac{1}{|X|} \sum_{x \in X} \max_{i < |x|} L(f(x_{0...i-1}), x_i) \quad (1)$$

In Fig. 4 *(a)(b)*, we plot this continuous metric for addition models on the length generalization dataset. The distribution of this metric across random seeds is still clustered. We therefore confirm that *emergent capabilities exhibit bimodal distributions even when using a continuous performance metric*; their bimodality is not due to thresholding alone. Further findings from these experiments are in Appx. D. In Appx. C.2, we also discuss that in count task, we observe inverse scaling in mean accuracy. The average of *successful* runs remains monotonic in these ranges, suggesting that U-shaped curves in overall mean are an artifact of success probability.

## 2.5. Bimodality Emerges Abruptly

The previous section showed that bimodal performance distributions produce emergent scaling curves for *individual seeds*. We next show that the *distribution itself* can change suddenly at certain scales. As shown in Fig. 2, the performance distribution starts as unimodal at the smallest scale, where no models can length generalize. Larger scales yield a bimodal distribution where most of the probability mass is ultimately placed on successful (i.e., length-generalizing) runs. A priori, there could be a smooth evolution between these two distributions in which probability mass from failing runs gradually shifts towards higher performance metrics, eventually splitting into a clear separate cluster. We find instead that the shift from unimodal to bimodal is abrupt and instantly polarized into low and high clusters. In Appx. Fig. 13, we confirm that models fail to generalize when fixing depth to be one

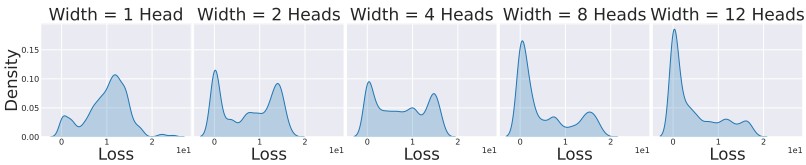

*(a)* Fixing network depth to 6 layers while scaling up width.

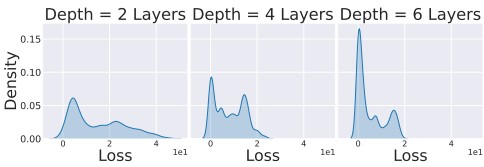

*(b)* Fixing network width to 8 heads per layer while scaling up depth.

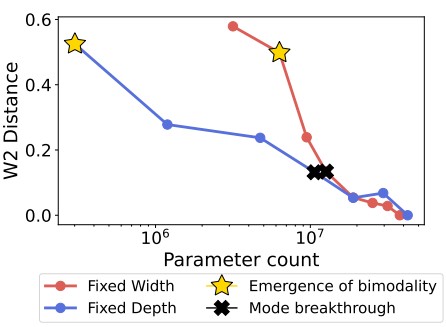

*(c)* Changes in random variation.

*Figure 4.* **Analysis of performance distributions (addition task). Left (a, b):** Kernel Density Estimation (KDE) of our loss-based error metric (Eq. 1). At scales where accuracy is bimodal, the distribution remains bimodal even with a continuous metric. **Right (c):** Wasserstein-L2 distance of each scale's performance relative to the largest scale. We mark the "emergence of bimodality" and the "mode breakthrough." During breakthrough, distributions shift gradually.

layer even when scaling width (despite having perfect in-distribution accuracy), illustrating that the length-general function is suddenly learnable at sufficient depth.

To track the evolution of the performance distribution across scales, we plot the Wasserstein-L2 distance of each scale's distribution relative to that of the final, largest model scale. In Fig. 4c we see a sharp decrease in the W2 distance, marking the sudden appearance of highly successful runs when model depth reaches 3 layers or model width reaches 2 heads. These sudden changes identify the moment when a new capability is unlocked, as the distribution transitions abruptly from unimodal to bimodal. We posit that this transition marks the **minimum capacity** required to learn the task. We also mark the point in each trend where the mode in Fig. 3 (top left) increases sharply. We argue that one can draw misleading conclusions about minimal model capacity when only studying using single runs at each scale, whereas *distributional statistics correctly identify the minimum capacity that a skill requires*.

## 3. Large Language Model Experiments

After connecting emergence with bimodality in small synthetic settings, we turn to LMs. We focus on the MMLU multiple-choice question answering dataset, where performance jumps (Srivastava et al., 2023) after the LM learns the multiple choice format (Hu & Frank, 2024). We will show that emergent scaling curves in LMs express underlying multimodal performance distributions.

### 3.1. Data and Setup

To avoid the expense of repeatedly training large LMs from scratch, we simulate independent runs by reinitializing the upper layer of pretrained LMs before continuing to train

them. Our continued training dataset, a mix of C4 news data and MMLU training data. MMLU's default training set (auxiliary train) contains a question domain distribution different from the test set. Furthermore, MMLU (Hendrycks et al., 2021) has a much larger test set, containing 14K questions as well as a smaller validation set, constaining 1.5K question, both drawn fromthe same domain distribution. Therefore, we use the official MMLU test set as the training data and refer to as the MMLU training data and the 1.5K validation questions as the held-out test data. We hypothesize that for compute scales near the emergence threshold, performance distributions will be bimodal across random seeds.

During pretraining, a large diverse corpus encourages models to acquire various capabilities. While sufficiently large models may learn all capabilities, smaller models have limited capacity, requiring capabilities to compete (Merrill et al., 2023). This competition, influenced by initialization and data order, leads to varying outcomes across random seeds, forming performance clusters.

**Task**: We test LMs on MMLU validation set (Hendrycks et al., 2021). Strong MMLU performance requires (1) natural language reasoning with domain knowledge and (2) producing answers in the required format. The latter drives emergent trends (Srivastava et al., 2023; Hu & Frank, 2024).

**Model**: We use the Qwen2.5 family of base models (Yang et al., 2024). To introduce randomness, we reinitialize the final attention layer and LM head, then perform full-parameter continued pretraining.

**Data**: We mix the C4 news subset (Raffel et al., 2023) with MMLU training data to ensure the multiple choice formatting circuits compete with general language modeling. We vary the MMLU proportion to control task-specific data size.

**Training**: We continue pretraining Qwen2.5-0.5B and Qwen2.5-1.5B on C4-MMLU mixes, training 60 reinitializations per data mixture ratio. We train for 2 epochs (0.5B) and 5 epochs (1.5B) with learning rate 1e-5 and linear decay. Overall, we noticed that the emergence of bi-modality is sensitive to batch size. For Qwen2.5-0.5B, we train with batch size 64. For Qwen2.5-1.5B, we train with batch size 8, and we did not obserce strong bi-modality with batch size 64.

### 3.2. Experiment Results

**Emergence Across Data Compositions.** When training data contains more MMLU examples, the Qwen2.5-0.5B performance distribution improves in ways that mirror the effect of scale. In Figure 5 (*top*), when trained on 5% MMLU and 95% C4 (Raffel et al., 2023), most models (out of 60 seeds) achieve near 0% MMLU accuracy, failing to process the multiple-choice format. At 7.5% MMLU training data, bimodality emerges: one cluster remains near 0% accuracy (format failure), while the other reaches ≈25% accuracy (random baseline). This second cluster contains models that consistently produce valid multiple choice responses. We did not observe any seeds exceeds far beyond 25% (trivial performance), indicating that all seeds have only learned to compose format following without recovering world knowledge. Due to the limited scale and quality of our continued pretraining data, fully recovering the base model's MMLU capability is challenging. At $> 14\%$ MMLU, all 60 seeds consistently perform around the random baseline. Qwen2.5-1.5B shows the same bimodality (Figure 5, *bottom*), but the larger model recovers similar capabilities at much lower data mixture ratios. Furthermore, the mean and median performance across random seeds for Qwen2.5-0.5B model shows emergence at 7.5% MMLU training mix ratio (Figure 5, *top left*), which corresponds to the emergence of bi-modality across seeds. In contrast, even the mean and median for Qwen2.5-1.5B (Figure 5, *bottom left*) shows linear progression, *concealing* the underlying bi-modality at 0.08% MMLU ratio.

**Emergence Across Model Scales.** Model size affects the clustered MMLU performance distribution similarly to the synthetic settings in Sec. 2. Using 10% MMLU training data, we continually pretrain Qwen after reinitializing with 80 different seeds (Figure 6). While Qwen2.5-0.5B forms two distinct performance clusters, Qwen2.5-1.5B consistently performs around or above the random baseline. Larger models reliably acquire MMLU capability on the same dataset that produces highly bimodal variation at smaller scales, consistent with scaling laws: smaller models require more training examples to match larger model performance (Rosenfeld et al., 2019; Kaplan et al., 2020).

### 3.3. Is MMLU emergence a mirage?

Is multiple choice performance only bimodal because accuracy is thresholded? As in our synthetic setting (Sec. 2.4), we examine continuous metrics. Specifically, instead of measuring the exact match accuracy on the correct answer choice, we measure the negative log likelihood on the correct answer choice token. In Fig. 9, we confirm that negative log likelihood (NLL) loss remains bimodal. For the 10% MMLU training data case, we confirm this bimodality is statistically significant ($p < 0.001$) for significant using Hartigan's Dip Test (Hartigan & Hartigan, 1985), which measures the maximum difference between the empirical distribution and the best-fitting unimodal distribution. The bimodality observed in NLL also mirrors our synthetic algorithmic task findings.

While recent work (Srivastava et al., 2023; Schaeffer et al., 2024) argues that discontinuities disappear with continuous metrics, Fig. 19 and Fig. 20 reveals the exact opposite: NLL loss exposes clusters *concealed* by accuracy metrics, particularly among low-performance models trained on 5% MMLU (Fig. 19) and among larger models (Fig. 20).

Is this clustering specific to emergent tasks? Fig. 10 suggests so. Pretraining loss produces a smooth, unimodal, nearly symmetric distribution, contrasting sharply with MMLU's multimodality. We therefore conclude that multimodality is linked to emergent tasks. It is *not merely* a downstream effect of in-distribution training noise.

## 4. Grammatical Generalization in LMs

We next examine English syntax acquisition during LM training. Question formation (QF) is a canonical benchmark (Mueller & Linzen, 2023; McCoy et al., 2020b;a) for grammatical rule learning. In this setting, the ambiguous training data supports two possible competing rules: (i) a hierarchical syntactic rule (correct for English) or (ii) a superficial linear heuristic (like an n-gram language model). On the disambiguating OOD test data, the hierarchical rule yields 100% accuracy, the linear rule yields 0%, and any score in between indicates inconsistent rule application.

Prior work has documented performance clustering in QF across random seeds (Qin et al., 2024; Ahuja et al., 2024; Murty et al., 2023), with evidence that data composition and model scale influence which rule is learned (Qin et al., 2024; Ahuja et al., 2024). This controlled setting allows us to study bimodal distributions of competing viable solutions when training a simple language model from scratch.

### 4.1. Data and Setup

**Task**: We pretrain models on the **question formation** task (McCoy et al., 2018), which tests English syntax acquisition.

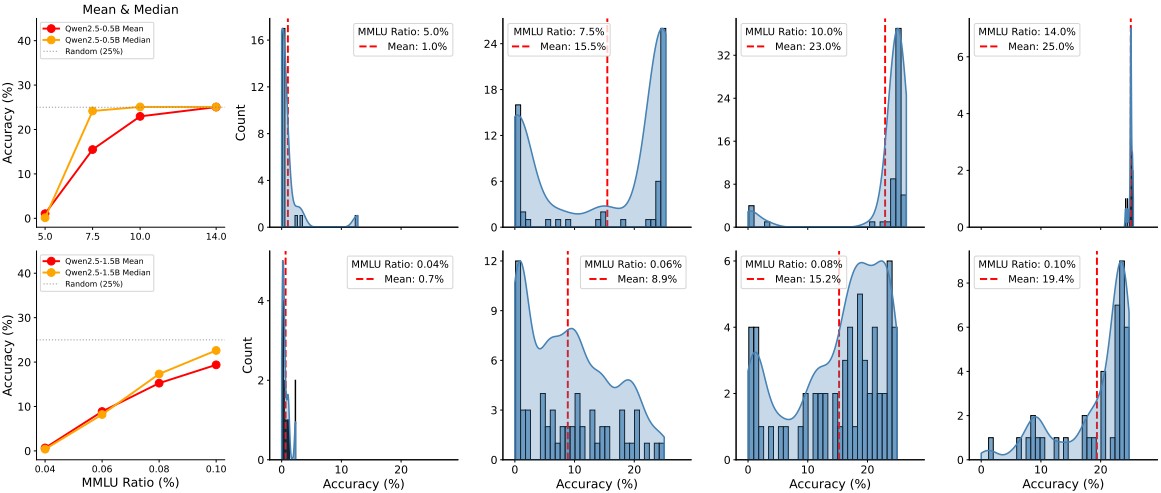

*Figure 5.* **Performance on MMLU across data compositions over 60 random seeds.** All models (*top:* Qwen2.5-0.5B, *bottom:* Qwen2.5-1.5B) have their last layer reinitialized randomly and then are trained on a mix of C4 news and MMLU, with varying proportions of MMLU in the training mix. With insufficient MMLU training data, performance is trivial; as data increases, some seeds achieve non-trivial performances, with Qwen2.5-0.5B overall requires a larger MMLU mix ratio than Qwen2.5-1.5B. Importantly, bi-mdality emerges in both cases.

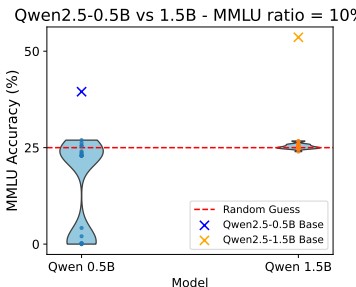

*Figure 6.* **Performance on MMLU across model scale over 60 random seeds.** For a fixed data mix (7.5% MMLU in training mix), smaller models show bimodal performance while larger models consistently relearn the task, albeit below original base levels. Each dot represents one seed.

Models transform declarative sentences to questions by moving the main auxiliary verb (e.g., "My unicorn *does* move" → "*Does* my unicorn move?"). The training data contains some sentences that induce the hierarchical rule (through *center embedding* syntactic structure) and some sentences that induce the linear rule (through *right-branching* syntactic structure). Critically, the question formation examples in the training data are *always ambiguous* between both rules. The OOD test data then probes for which rule has been learned. (Full task details in Appx. E.)

**Model**: Decoder-only Transformers trained from scratch on causal language modeling. Following prior work (Qin et al., 2024; Murty et al., 2023), we vary depth (3-8 layers) with 8 heads per layer and 512-dimensional embeddings.

**Data**: Synthetic English sentences generated via Context-Free Grammars (McCoy et al., 2018). Based on Qin et al.

(2024), we vary the proportion of hierarchical-inducing versus linear-inducing sentences in the training data, paralleling our MMLU data mixture experiments.

**Training**: We train 80 random seeds per configuration. We use Adam optimizer with learning rate $1 \times 10^{-4}$ and linear decay and train models for 600K steps. We vary (1) model size and (2) proportion of hierarchy-inducing training examples, to study how these scale factors influence the bimodal performance distribution of hierarchical generalization.

### 4.2. Experiment Results

**Emergence Across Data Compositions.** Fig. 7 (*left*) shows mean OOD accuracy exhibits sharp emergence at 10% hierarchical data, jumping from near 0% to 80%. However, histograms (*right 4 panels*) reveal that the population mean masks the underlying bimodality: at 10%, seeds split between 0% (linear rule) and 80-100% (hierarchical rule). Varying data composition shifts *which* rule models learn. More hierarchical-inducing sentences increase the probability of learning the hierarchical rule, but do not eliminate the competing solution even at 100% hierarchical-inducing data. Apparently, even our most hierarchical-inducing training set offers insufficient support to provide consistently "successful" runs, possibly because it lacks diversity.

**Emergence Across Model Scales.** Fixing the data mixture at 90% hierarchical sentences, we now examine how model capacity affects the performance distribution. At 3 layers (Fig. 8), performance is unimodal around 70%, with few seeds reaching 100%—these smaller models lack capacity to learn either systematic syntactic rule. At 6 and 8 layers, clear bimodality emerges: most seeds achieve

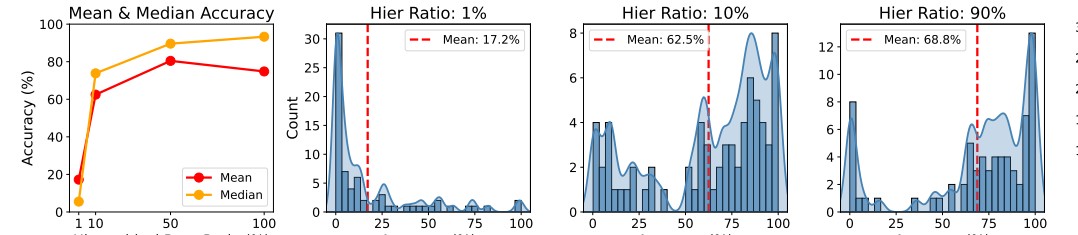

*Figure 7.* **Question formation performance across data compositions.** *Left*: Mean and median OOD accuracy across 80 seeds shows emergence at 10% hierarchical-rule inducing data. *Right-4*: Histograms reveal bimodal distributions at 10%, 90% and 100% hierarchical data. Models either cluster at either 0% (linear generalization) or 100% (hierarchical generalization).

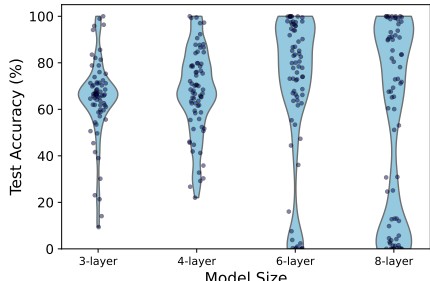

*Figure 8.* **Question formation performance across model scales.** Training on 90% hierarchical-rule inducing data, 3-layer models show unimodal distribution around 70% OOD accuracy, while 6-layer and 8-layer models exhibit bimodality with most seeds achieving 100% (hierarchical generalization) and some reaching 0% (linear generalization). Each dot represents one seed.

100% (hierarchical generalization) while some reach 0% (linear generalization). Although our largest scale models tend to consistently apply systematic rules, we find different seeds still vary in *which* rule they apply. Finally, similar to Sec. 3.3, we confirm that the bi-modality is still observed under continuous metric in Fig. 9 and is statistically significant under Hartigan's Dip Test (Hartigan & Hartigan, 1985).

## 5. Conclusion & Discussions

Our work explores the evolution of random variation in model performance across scales, bringing a nuanced perspective on emergent capabilities. While the mode of "emergent" performance distributions may sharply improve at a certain model scale, we attribute these sudden jumps to gradual improvements in the random distribution. In fact, bimodality often emerges *before* the mode—or most individual runs—exhibits a breakthrough, even when the transition from a unimodal distribution to a bimodal one is sudden.

### 5.1. Discussions

**Random variation:** Model performance can be sensitive to stochastic aspects of the training process like random initialization and training data order. While variation may

have a benign effect on in-distribution performance (Jordan, 2024), previous works indicate that out-of-distribution performance can fluctuate substantially across training runs, even at larger scales (Madaan et al., 2024). Prior studies have documented performance differences across various stress test sets (D'Amour et al., 2022; Naik et al., 2018), including length generalization (Zhou et al., 2024b;a). More generally, out-of-distribution behavior like compositional rules (McCoy et al., 2019) or associative biases (Sellam et al., 2022) often exhibit extreme variation compared to in-distribution loss. Such differences persist throughout training, not just at the final checkpoint (Zhou et al., 2020). Dodge et al. (2020) compared the impacts of weight initialization and data ordering, concluding that both contribute equally to variation in performance.

Existing work has also found model runs can cluster in shortcut learning (Juneja et al., 2023; Li et al., 2025) and in training dynamics (Qin et al., 2024; Hu et al., 2023), hinting at multimodal variation. Our work connects these random clustering effects to the phenomenon of emergence at scale. Relevant to our setting, Zhou et al. (2024a) and Zhou et al. (2024b) provided evidence of variability in length generalization across random seeds, which we further analyze across model scales. We also expand on the findings of Qin et al. (2024) of bimodal variation in LM syntax acquisition.

**Emergent abilities of LMs:** In LMs, scaling laws predict reliable performance improvements as models increase in size or train on larger datasets (Hestness et al., 2017; Rosenfeld et al., 2019; Brown et al., 2020; Kaplan et al., 2020). Emergent abilities are abilities that arise unexpectedly, out of line with these predictions (Srivastava et al., 2023; Ganguli et al., 2022; Wei et al., 2022). These abilities are characterized by unpredictable and abrupt performance improvements on specific benchmarks at certain scales. Although recent studies suggest that some breakthroughs may stem from the choice of evaluation metrics rather than fundamental changes in model behavior (Srivastava et al., 2023; Schaeffer et al., 2024), other breakthrough capabilities remain emergent—8% of high-breakthroughness datasets in BIG-Bench, according to Schaeffer et al. (2024). Our find-

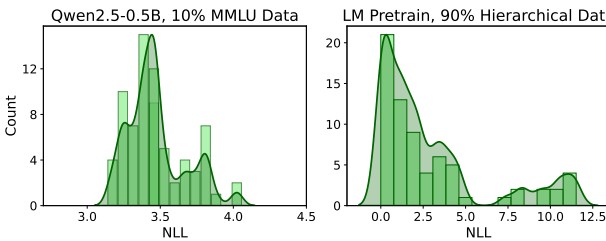

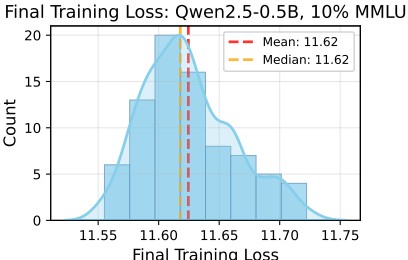

*Figure 9.* **NLL loss distributions.** Even with a **continuous** metric (NLL), we still observe multi-modal distributions in performance across random seeds. NLL is measured on: *Left:* MMLU for Qwen2.5-0.5B trained on 10% MMLU mixture. *Right:* OOD QF for LM pretrained on 90% hierarchical mixture.

*Figure 10.* **KDE of LM training loss.** The distribution across 80 seeds is unimodal and smooth. By contrast, Fig. 9 depicts MMLU loss as irregularly clustered.

ings suggest that careful statistical analysis is required to evaluate claims both for and against "emergence" scaling on specific tasks.

Snell et al. (2024) found that some scales exhibit earlier emergence if finetuned explicitly on an emergent task, suggesting that smaller models may have the capacity for that task but are limited by its scarcity in the training corpus. Similarly, we show that emergent capabilities can arise from multimodal random variation using synthetic length generalization tasks as a case study.

**Depth versus Width Scaling:** Downstream performance varies with architecture shape, not just model size (Tay et al., 2022). For compositional tasks, deeper models often generalize better, but for fixed compute budgets, shallower and wider models may be advantageous (Petty et al., 2024). Several explanations of this benefit have been proposed. Edelman et al. (2024) argued that wider networks offer more parallel queries over randomized subnetworks and therefore learn sparse features efficiently. Levine et al. (2020) established width-dependent depth thresholds beyond which depth yields diminishing returns (Levine et al., 2020). We investigate how independently scaling width and depth influences random variation in compositional tasks, documenting a surprising regime (Appx. C.2) where increasing width damages performance while increasing depth improves it. Our findings should inspire further study of how emergent tasks respond to architectural hyperparameter tradeoffs.

### 5.2. Limitations

**Computational and experimental constraints.** Ideally, in our LLM experiments (Sec. 3.1) we would pretrain full language models from scratch across many model scales and many random seeds, but end-to-end pretraining for 0.5B and 1.5B models across 80 seeds can be prohibitively expensive. Instead, our LM experiments rely on continual pretraining partially reinitialized Qwen2.5 models. The reinitialization introduces a specific form of randomness that differs from training entirely from scratch. As a result, our set up could either amplify or suppress bimodality relative to the true from-scratch pretraining setting.

Furthermore, we mostly recovered the multiple-choice formatting ability, while full language modeling capabilities are difficult to fully recover given the size of our continual pretraining set (m̃illions of tokens) compared to the Qwen pretraining corpus ( potentially billions or trillions of tokens). This resource gap also explains why we do not examine reasoning or compositional tasks like GSM8K.

**Sensitivity to training choices.** In Sec. 4, we have observed that bi-modaity can be sensitive to hyperparameter choices. However, our analysis does not fully characterize how bimodality might be sensitive to hyperparameter choices such as learning rate, batch size, or optimizer. If bimodality proves fragile to these choices, the practical implications of our framework for scaling predictions would be narrower than suggested.

**Theory and mechanistic understanding.** Although we document when and where bimodal performance distributions appear, we do not provide a formal theoretical framework for when or why bimodality must arise. We also do not fully characterize the mechanistic origin of the phenomenon: for example, how optimization landscapes, data composition, or competition between circuits give rise to distinct performance clusters. Developing formal conditions for the emergence of bimodality and understanding its causal mechanisms are important directions for future work.

### Impact Statement

Our findings have implications for resource allocation in model training: practitioners should consider training multiple seeds at moderate scales rather than single runs at larger scales, particularly for tasks exhibiting emergent behavior. Our results also suggest that architectural choices and data composition can shift which algorithmic solutions models discover, highlighting the importance of careful experimental design beyond simply scaling compute. Understanding these stochastic dynamics is crucial for making AI systems more reliable and predictable as they continue to scale.

## Acknowledgments

This work has been made possible in part by a gift from the Chan Zuckerberg Initiative Foundation to establish the Kempner Institute for the Study of Natural and Artificial Intelligence. RZ, TQ, and SK acknowledge support from the Office of Naval Research under award N00014-22-1-2377 and the National Science Foundation Grant under award #IIS 2229881. RZ is supported by a Kempner Institute Graduate Research Fellowship, Simons Investigator Fellowship, NSF grant DMS-2134157, DARPA grant W911NF2010021,and DOE grant DE-SC0022199. TQ and DAM are partially supported by the Kempner Institute, the Aramont Fellowship Fund, and the FAS Dean's Competitive Fund for Promising Scholarship. Our work has been improved by invaluable discussion with Will Merrill and David Chiang.

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

# A. Experimental details on synthetic tasks

Below we provide more details on the **count** and **addition** task settings. Hyperparameters for our decoder-only Transformer models are largely adapted from (Zhou et al., 2024a). We train all of our models to convergence on the train distribution. Our experiments were run on an internal cluster and all model scales can be run on a single 40GB A100 GPU with gradient accumulation; for **count**, runs can finish within 2 hours and for **addition**, runs can finish within 6 hours. At each scale, we train models from 250 seeds for the count task and 200 seeds for reverse order addition.

**Count task:** For all of our training runs, we fix the vocabulary size to 150. For thresholded evaluation, we compute the exact match (EM) accuracy across all consecutive subsequences of the test length.

- **Model scales:** As mentioned in Sec. 2.1, we scale up our models by fixing width and scaling depth and fixing depth and scaling width. The precise parameters for each variation are as follows. For our **fixed depth experiments**, we fix the network depth to 4 layers and vary width by taking hidden dimensions $\{64, 128, 256, 384, 512, 640, 768, 1024\}$. The head dimension is fixed to 64. For our **fixed width experiments**, we fix the hidden dimension to be 512 and vary the depth from $\{1, 2, 4, 6, 8\}$ layers.
- **Hyperparameters:** We use a learning rate of $1e-3$ with a cosine decay scheduler and weight decay 0.1. We set the maximum training duration to be 10000 steps, with batch size 128 and context length 256.

**Reverse Order Addition with Index Hints:** For thresholded evaluation, we compute the exact match (EM) accuracy across 500 batches of 128 examples each.

- **Model scales:** For our **fixed depth experiments**, we fix the network depth to 6 layers and vary width by taking hidden dimensions $\{64, 128, 256, 384, 512, 640, 768\}$. For our **fixed width experiments**, we fix the hidden dimension to be 512 and vary the depth from $\{1, 2, 3, 4, 6, 8, 10, 12\}$ layers.
- **Hyperparameters:** We use a learning rate of $1e-4$ with a cosine decay scheduler and weight decay 0. We set the maximum training duration to be 30000 steps, with batch size 64 and context length 512.

# B. Breakthroughness and Linearity

Srivastava et al. (2023) introduced *breakthroughness* and *linearity* metrics to capture model performance improving suddenly or reliably with scale. Given a model's performances $y_i$ at model scales $x_i$ sorted by ascending model scale, the linearity metric $L$ and breakthroughness metric $B$ are respectively calculated as

$$L = \frac{I(y)}{\text{RootMeanSquare}(\{y_{i+1} - y_i\}_i)} \tag{2}$$

$$B = \frac{I(y)}{\text{RootMedianSquare}(\{y_{i+1} - y_i\}_i)} \tag{3}$$

where $I(y) = \text{sign}(\arg\max_i y_i - \arg\min_i y_i)(\max_i y_i - \min_i y_i)$.

In Fig. 11 we sample the five top seeds for the breakthroughness and linearity metric respectively for (a) count and (b) addition.

# C. Count task

Given two numbers in increasing order, the model is trained to generate a sequence which counts consecutively from the first number to the second number. Examples are given in the form `"5, 9 >, 5, 6, 7, 8, 9"`, while limiting the length of the counting sequence during training.

Zhou et al. (2024a) showed that models trained to count can generalize to more than twice this training length; however, Appx. C.2 reveals a more nuanced view of length generalization based on its distribution across independent model runs.

### C.1. Count performance distributions

Our plots in the main paper show distributions of addition task accuracy with a fixed test length 40 (Fig. 2). In Fig. 12 we show analogous histograms for the count task with a fixed test length of 60. We note that unlike addition, the mode corresponding to failing runs disappears at the highest widths and depths ran, indicating a difference in task difficulty

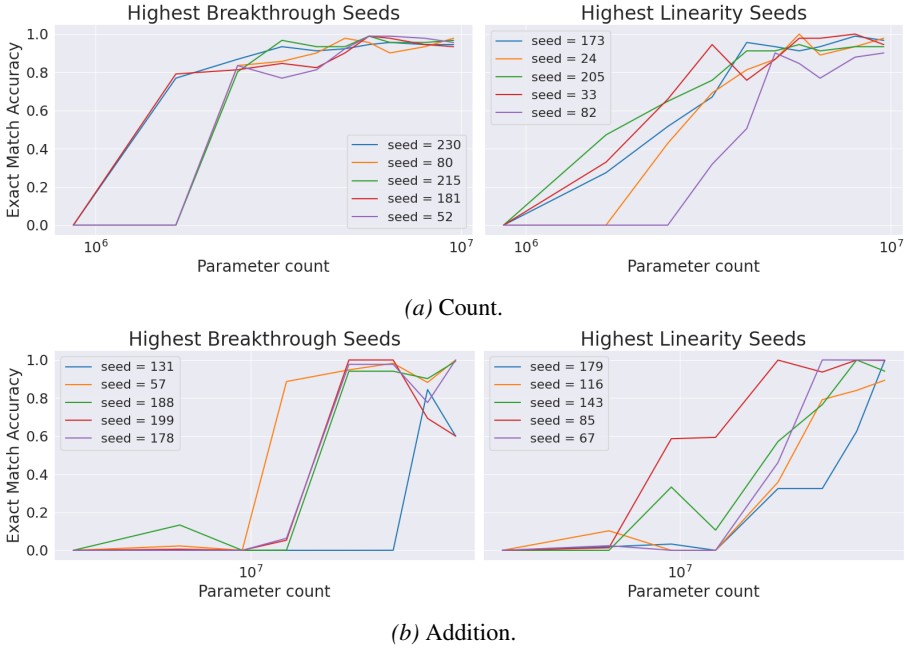

*(a)* Count.

*(b)* Addition.

*Figure 11.* **Top five seeds according to breakthroughness and linearity metrics.** Definitions for the two metrics are given in Appx. B, with resulting seeds plotted for a count and b addition.

affecting the random variation distribution.

In Sec. 2.5, we posit that the performance distribution becomes bimodal at the **minimum capacity** required by the task. For example, Fig. 13 shows that, regardless of width, models with a fixed depth of 1 layer are unable to length generalize on the count task, despite achieving near-perfect accuracy in-distribution.

### C.2. Count task as a case of U-shaped random scaling

We next consider the counting task (Sec. 2.1), which also exhibits bimodally-distributed performance (see Appx. Fig. 12) but yields a very different scaling effect: a U-shaped curve. Fig. 15 reveals this peculiar phenomenon in the mean accuracy scaling curve, when holding depth fixed. This curve is not simply a result of the summary statistic chosen, as it is mirrored by the evolution of the distributions as a whole according to their W2 distance in Fig. 14.

U-shaped scaling has been observed in LMs, but its causes are not currently well-understood. When the Inverse Scaling Prize (McKenzie et al., 2022) solicited tasks which exhibit inverse scaling trends—performance decreasing with scale—for large models, Wei et al. (2023) revealed that the majority of awarded tasks actually exhibit U-shaped scaling after considering even larger models. Treating the counting task as a concrete instance of U-shaped scaling at small model scales, we find that this unusual trend is still underscored by monotonic continuous changes in the performance distribution. Indeed, Fig. 15 (bottom right) shows that although the trend in the mean across all runs is U-shaped curve, the mean of the "successful" runs—those achieving at least 50% accuracy—still improves monotonically when increasing width. The observation of inverse scaling is, instead, due to changes in the *probability* of success (bottom left). Even when inverse scaling is in effect across a performance distribution, the performance of successful runs may exhibit more conventional responses to scale.

## D. Additional Continuous Metric Plots

**Addition Task.** In Sec. 2.4 we have seen seen that the addition histograms are bimodal in exact match accuracy (Fig. 2) and confirmed that the bimodal random variation distribution persists when viewing a continuous performance metric (Fig. 4 *a, b* ). We can consider another continuous metric, which corresponds to the minimum probability assigned to a token for each sample in the test length generalization dataset:

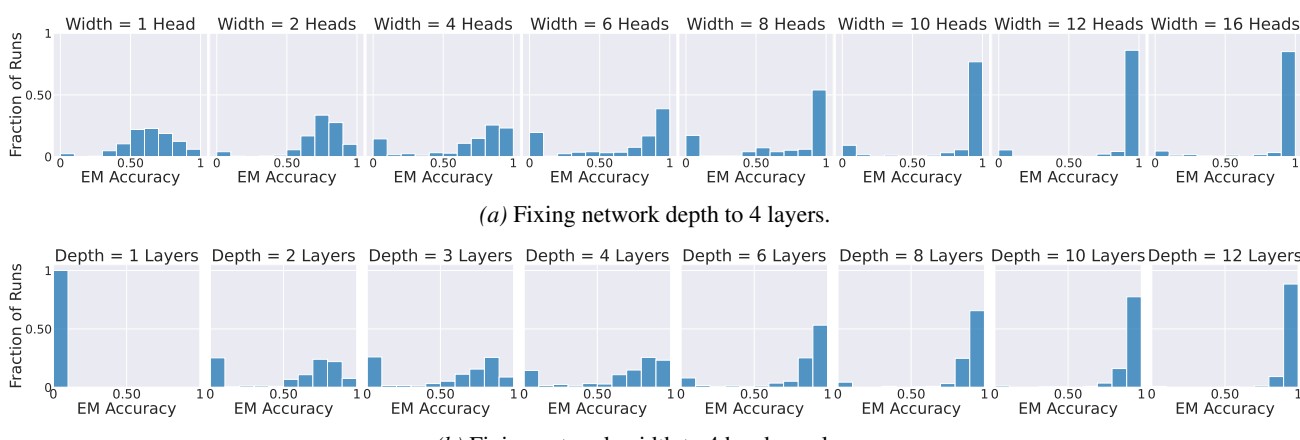

*(a)* Fixing network depth to 4 layers.

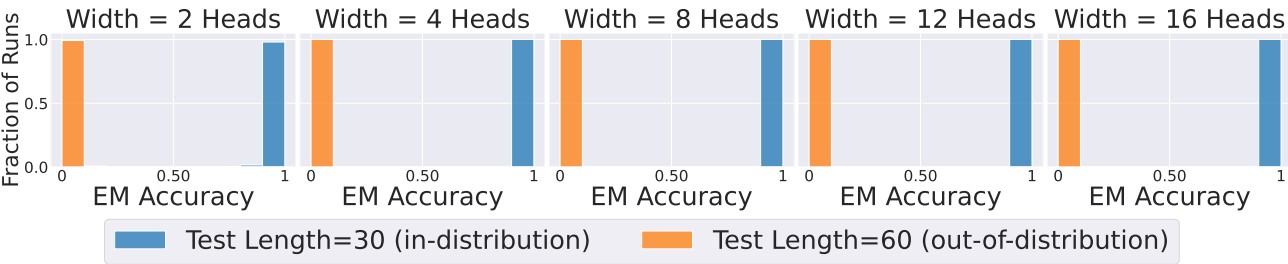

*(b)* Fixing network width to 4 heads per layer.

*Figure 12.* **Random variation in length generalization (count task).** Histograms of exact match accuracy on length 60 sequences when independently scaling (a). width and (b). depth.

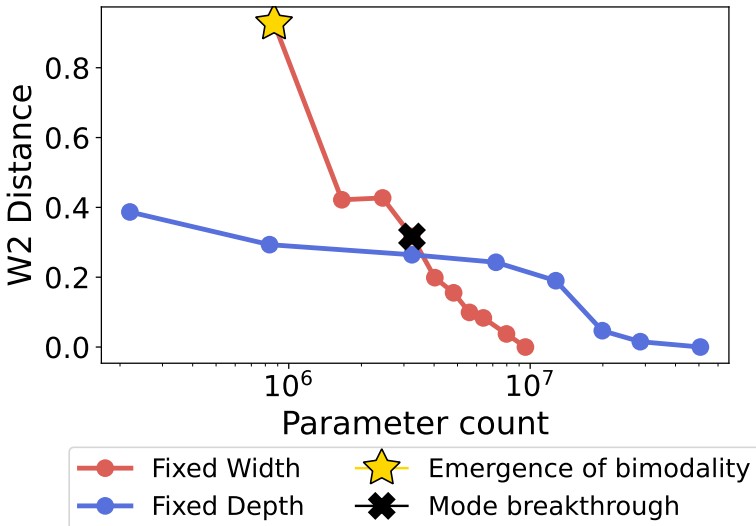

*Figure 13.* **Histograms for EM accuracy when fixing depth to be one layer (count task).** We plot the random variation distribution at test length 30 (blue, i.e. in-distribution) and test length 60 (orange, i.e. out-of-distribution). While all model seeds obtain near perfect accuracy in-distribution, all model seeds fail to length generalize at this depth.

*Figure 14.* **Changes in random variation (count task).** Wasserstein-L2 distance of each scale's performance distribution relative to the largest scale, scaling depth and width independently. We mark the emergence of bimodality at the last scale before multiple peaks appear. We mark the mode breakthrough at the last scale before successful length generalization becomes marginally more likely than failure. *The distribution changes slowly between intermediate model scales, but changes suddenly at the nadir of the U-shaped curve in Fig. 15.*

$$\text{minprob}(f, X) = \frac{1}{|X|} \sum_{x \in X} \min_{i < |x|} \mathbf{P}_f[x_i | x_{0 \dots i-1}] \tag{4}$$

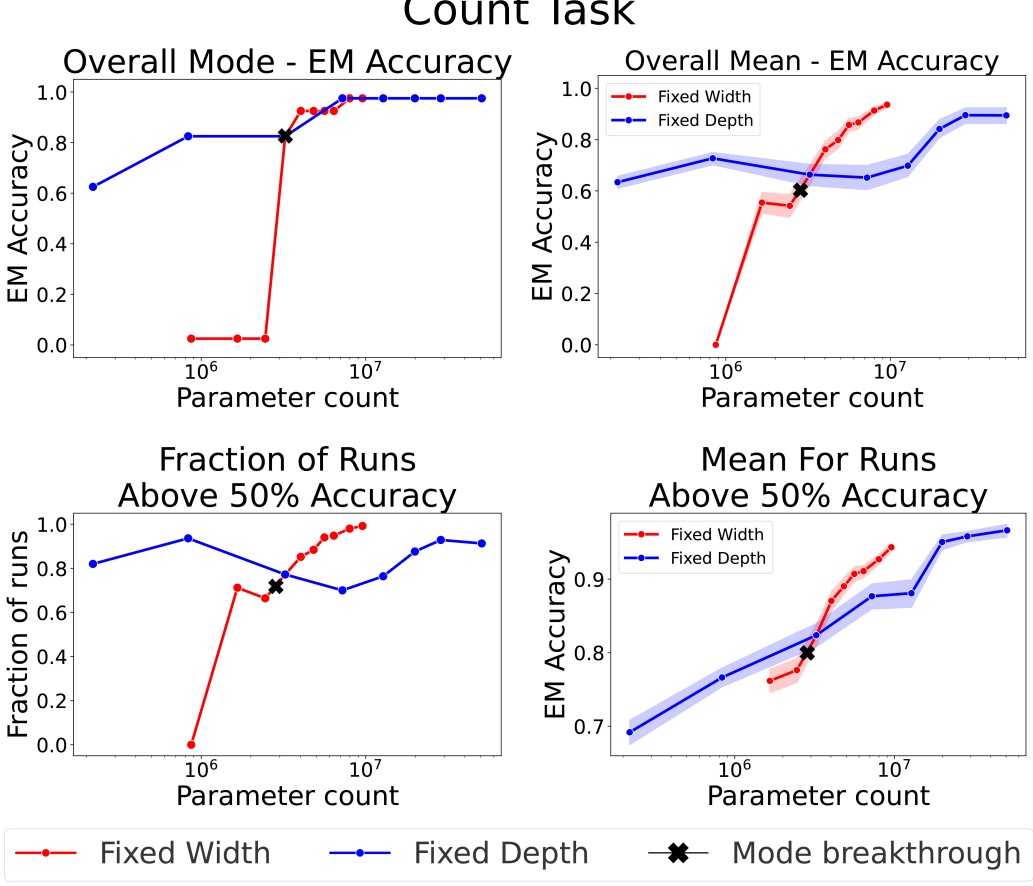

*Figure 15.* As in Fig. 3, we capture summary statistics for test length 60 on the count task. For fixed width (scaling depth), the trends across overall mode, overall mean, fraction, and mean of successful (above 50% accuracy) runs are similar to the reverse order addition case. The trend for fixed depth (scaling width) exhibits a U-shaped curve emerge for the mean across *all* runs; however, the mean of *successful* runs still exhibits continuous improvement.

We consider the average minimum probability (and analogously average maximum loss) in the sequence because errors are rare, and thus averaging across the sequence obscures generalization failures. However, as the model improves, the lowest-probability token may shift, but this transition is still continuous, not abrupt. Thus, unlike 0/1 accuracy, this metric also avoids threshold effects and better reflects gradual improvements in length generalization, similar to the continuous error metric. We plot the resulting histograms of this metric in Fig.s 16 and 17 for addition and count respectively. We note that bimodality persists for both tasks, particularly for addition; in particular, stronger bimodality exhibited in the accuracy histograms corresponds to stronger bimodality in the probability histograms.

**Count Task.** In Fig. 18, we plot the analogous KDE plot of the loss-based error metric for the count task. For count, we note that the mode corresponding to 'failing runs' when looking at accuracy or probability is more diffuse compared to addition. However, we claim that stronger bimodality in the original accuracy/probability plots in Fig.s 12 and 17 is still associated with stronger bimodality in the loss plots. For the addition task, the scales exhibiting the strongest bimodality in accuracy and probability are for width 4 and 8 (in the case of fixed depth) and depth 2 and 4 (in the case of fixed width), and these are also the most strongly bimodal loss KDE distributions. For the count task, even the most strongly bimodal setting (fixed width, with depth 2) has a low probability of failure ( 25%), but still produces a visible wide peak elevated over the long tail of the loss distribution.

**Large LM Experiments.** Finally, Fig. 19 provides a copy of Fig. 9 with both accuracy and NLL loss comparable for easy contrast. We note that high-performance clusters are distinguishable in both accuracy and loss, but the thresholded accuracy conceals low-performance clusters which are visible in the continuous distribution. We also include histogram version of

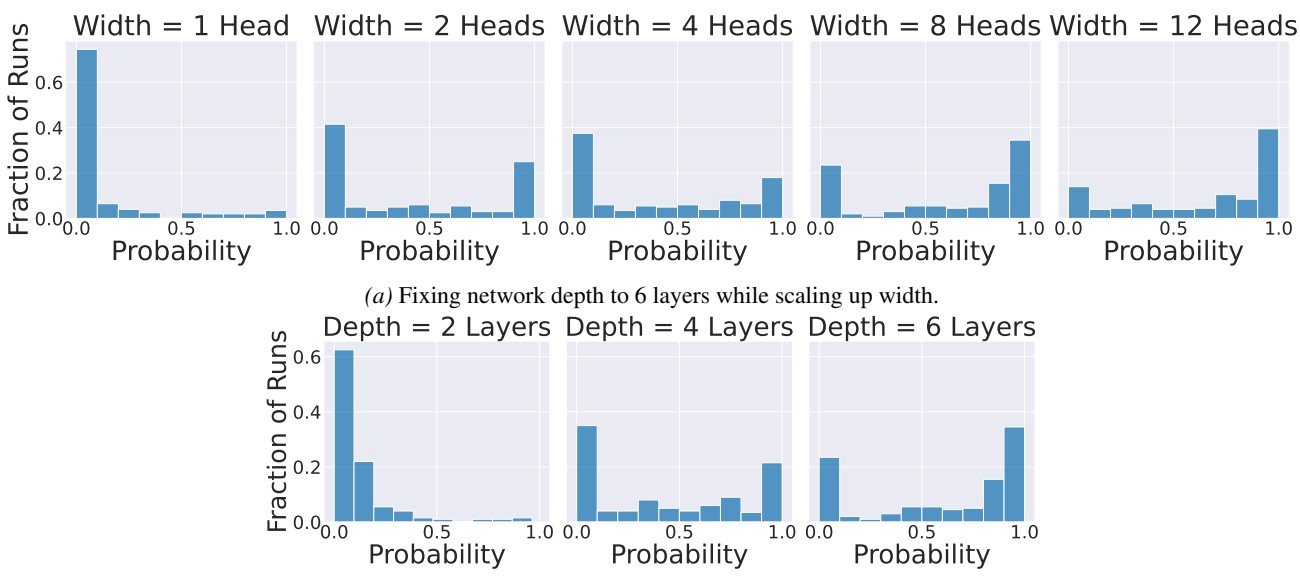

*(a)* Fixing network depth to 6 layers while scaling up width.

*(b)* Fixing network width to 8 heads per layer while scaling up depth.

*Figure 16.* **Histograms of the minimum probability of any token in each sequence, averaged across sequences (addition task).** Metrics are calculated by Equation 4. The bimodal nature of the random variation distribution persists even when using this continuous metric, analogous to Fig. 4 (a, b).

results in Fig. 20. Due to computation limit, for each data mixes, we train on 20 random seeds. Similar to Fig. 19. models yields the same bimodal distribution in accuracy metrics as well as multi-modal distribution in continuous metric (NLL).

## E. Grammatical Generalization Details

### E.1. Task Description

In the **question formation (QF)** task, models transform declarative English sentences into questions by moving an auxiliary verb to the front of the sentence. The task tests whether models learn the correct hierarchical syntactic rule based on the sentence's syntax tree structure, or instead rely on a superficial linear heuristic.

Our training data (based on McCoy et al. (2018)) permits two strategies for choosing which verb to move:

1. **Linear rule**: Move the first auxiliary verb in the sentence (incorrect for English)
2. **Hierarchical rule**: Move the main auxiliary verb, determined by the sentence's syntactic structure (correct for English)

Examples of each rule are provided in Table 1. The first example is **ambiguous** because both the hierarchical and linear rules produce the same correct output. In contrast, the second example is **unambiguous** because only the hierarchical rule produces the grammatically correct question.

### E.2. Data Composition

The training data contains two components:

1. **Question formation examples**: Transform declarative sentences to questions. These examples are always ambiguous between linear and hierarchical rules.
2. **Declaration copying examples**: Simply repeat declarative sentences without transformation. Unlike QF examples, these can include unambiguous sentence structures.

Following Qin et al. (2024), we manipulate the declaration copying subset to control exposure to different syntactic structures. Specifically, we vary the ratio of **center-embedded sentences** versus **right-branching sentences**:

- **Center-embedded sentences** contain recursive clause embedding (e.g., "My unicorn who doesn't sing does move"), which requires hierarchical syntactic processing and encourages learning the hierarchical rule.

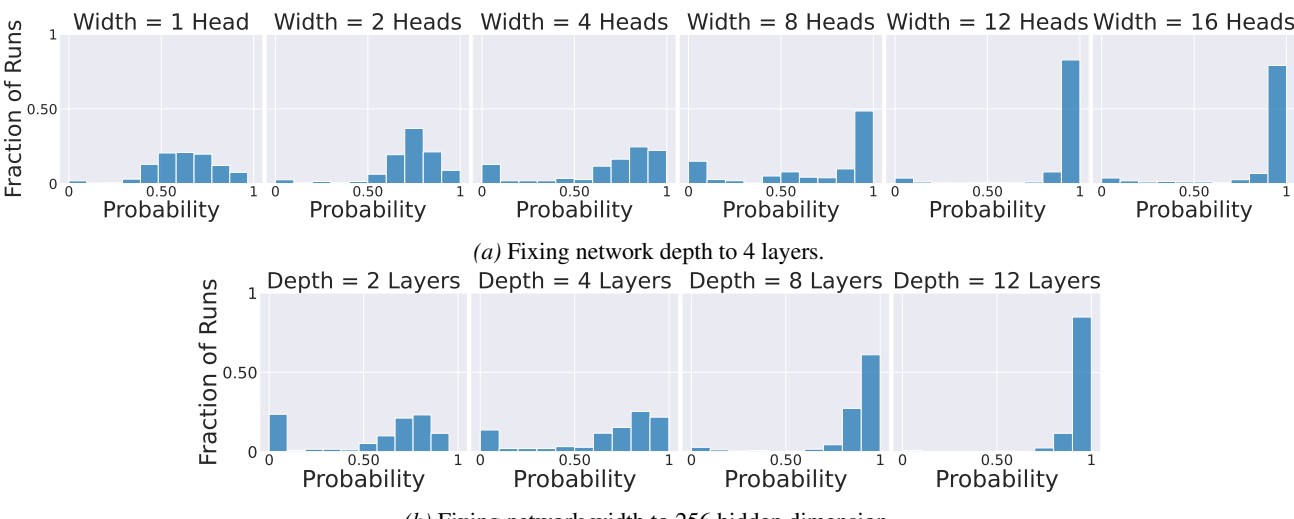

*(a)* Fixing network depth to 4 layers.

*(b)* Fixing network width to 256 hidden dimension.

*Figure 17.* **Histograms of the minimum probability of any token in each sequence, averaged across sequences (count task).** Random variation still leads to bimodal performance distributions, even using this continuous performance metric.

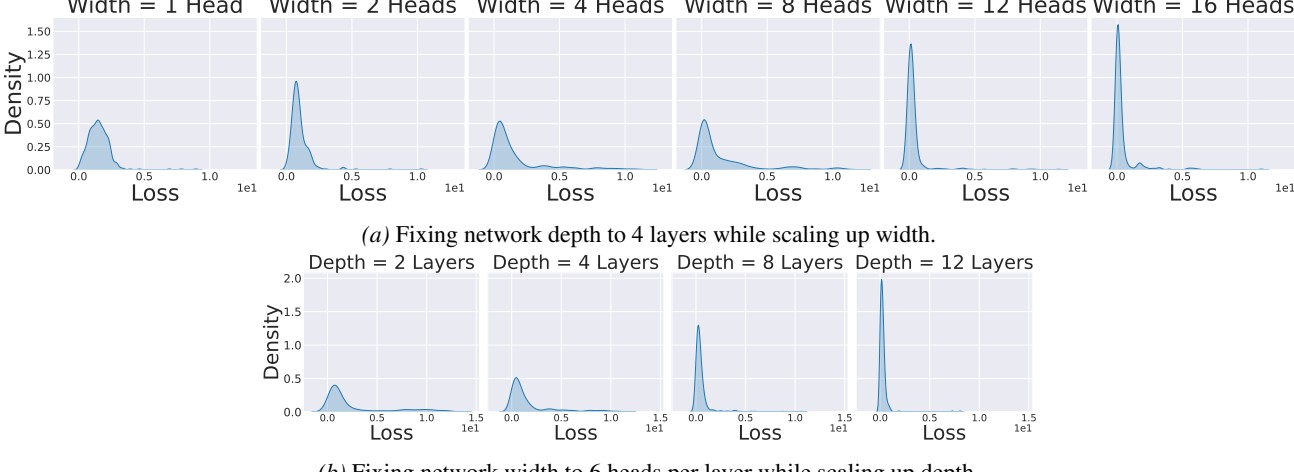

*(a)* Fixing network depth to 4 layers while scaling up width.

*(b)* Fixing network width to 6 heads per layer while scaling up depth.

*Figure 18.* **Random variation in continuous length generalization error (count task).** Kernel Density Estimation (KDE) of loss-based error metric (Equation 1) distribution across model runs. At scales where the EM accuracy distribution is most strongly bimodal (width=4, width=8, and depth=2), KDE places the most density at areas of high loss.

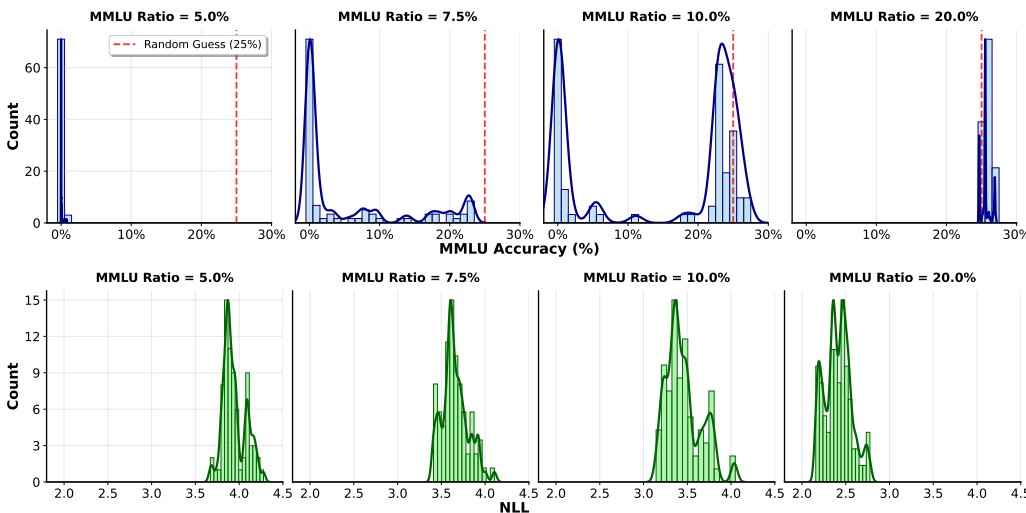

*Figure 19.* **KDE approximations of performance variation on the MMLU test set.** Provided for contrasting accuracy and continuous NLL loss. *Top:* Approximate distribution of accuracy metrics. *Bottom:* Approximate distribution of KDE metrics, copied from Fig. 9.

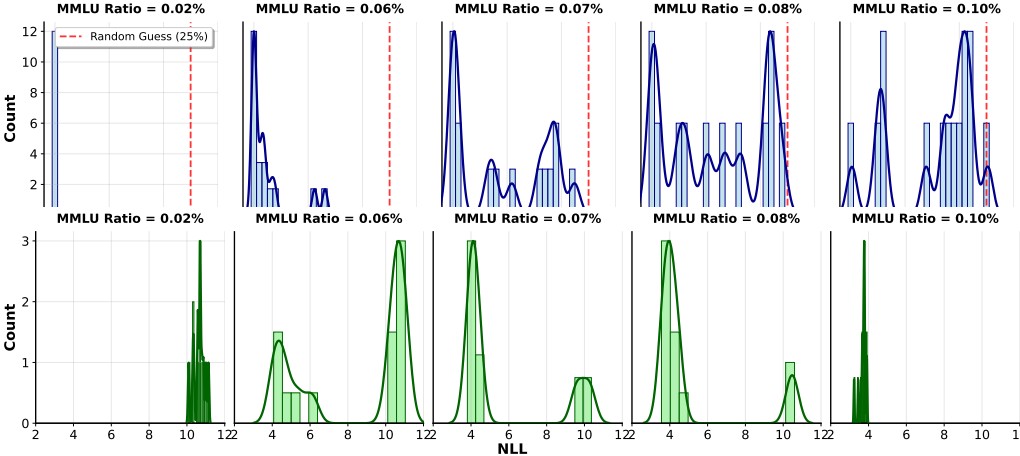

*Figure 20.* **Accuracy and NLL loss for Qwen2.5-1.5B MMLU experiments.** We train last-layer reinitialized Qwen2.5-1.5B on data mixes with different ratios of the target task (MMLU). *Top:* Distribution of accuracy metric. *Bottom:* Distribution of NLL loss metric.

- **Right-branching sentences** have simpler linear structure (e.g., "My unicorn does move the dogs that do wait"), which can be solved with linear n-gram patterns.

We create training sets with different mixtures of these sentence types in the declaration copying subset: *Quest Only* (no declaration copying), *Center Embed* (only center-embedded declarations), *Right Branch* (only right-branching declarations), and various intermediate mixtures.

### E.3. Evaluation

The in-distribution test set contains only ambiguous examples (solvable by both rules), while the OOD test set contains only unambiguous examples (requiring the hierarchical rule). Models using the hierarchical rule achieve ∼100% accuracy on both test sets. Models using the linear rule achieve ∼100% on in-distribution data but ∼0% on OOD data. We therefore use **OOD accuracy** as our measure of whether models have acquired the hierarchical syntactic rule.

*Table 1.* **Question Formation task examples.** Models must move the main auxiliary verb to the front to form a question. Ambiguous examples can be solved by either linear or hierarchical rules; unambiguous examples require the hierarchical rule.

| Task | Example Type | Examples |
|------|-------------|----------|
| Question Formation | Ambiguous | **Input:** My unicorn does move the dogs that do wait. |
| | | **Output:** Does my unicorn move the dogs that do wait? |
| | Unambiguous | **Input:** My unicorn who doesn't sing does move. |
| | | **Linear Output:** Doesn't my unicorn who sing does move? |
| | | **Hierarchical Output:** Does my unicorn who doesn't sing move? |

