# OpenReview forum: "Random Scaling of Emergent Capabilities"
_ICML.cc/2026/Conference — ICML 2026 regular_

### Official Review · Reviewer_CftN · 2026-02-17

**Soundness:** 3
**Presentation:** 3
**Significance:** 3
**Originality:** 3
**Overall Recommendation:** 5
**Confidence:** 4

**Summary:**

This paper examines why some language model capabilities appear to
emerge suddenly despite overall smooth scaling trends. Rather than
attributing breakthroughs to discrete capacity thresholds or metric
artifacts, the authors argue they arise from bimodal performance
distributions across random seeds.  As model scale increases, the
probability of successful training runs shifts continuously, even
though individual runs may show abrupt jumps.  Experiments on
synthetic tasks, MMLU, and grammatical generalization demonstrate that
both emergent and smooth scaling curves can result from seed
variation.  The study concludes that emergent capabilities reflect
probabilistic dynamics in training outcomes rather than true
discontinuities in model capacity.

**Compliance With Llm Reviewing Policy:**

Affirmed.

**Final Justification:**

The rebuttal resolved my concerns.

**Key Questions For Authors:**

-What conceptual and methodological challenges would arise in extending the proposed bimodality framework from single-level splits to multi-stage or hierarchical bifurcations of capabilities?

-Is unimodality at small scales a theoretical necessity of the proposed
framework, or merely an empirical observation within the tested seed
budget?

**Limitations:**

The paper does not include a dedicated Limitations section. While certain constraints—such as the use of controlled settings and continued pretraining rather than full-scale pretraining—are implicitly acknowledged, they are not systematically discussed. As a result, the scope and external validity of the proposed framework are not explicitly delineated.

**Strengths And Weaknesses:**

A key strength of this paper is the introduction of bimodality across
random seeds as a novel explanation for emergent scaling phenomena. By
shifting the focus from single-run performance curves to
population-level performance distributions, the authors provide a
probabilistic reinterpretation of emergence. This perspective is
conceptually original and supported by empirical evidence across
multiple tasks and metrics.

While the central idea of bimodality is compelling, the contribution
remains primarily empirical in nature. The paper does not provide a
formal theoretical framework explaining when or why bimodality must
arise. As a result, some of the claims (e.g., minimum capacity
thresholds) remain interpretive rather than rigorously
derived. However, this limitation does not diminish the value of the
work, as the extensive empirical evidence meaningfully advances the
discussion on emergent scaling phenomena.

The empirical analysis focuses primarily on single-level bimodal
splits (success vs. failure) within individual tasks. The paper does
not investigate whether emergence may involve multi-stage or
hierarchical bifurcations across interacting skills. As a result, the
generality of the proposed bimodality framework for more complex,
multi-level capability structures remains unclear.

---

> ### Author Rebuttal · Authors · 2026-03-30
>
> Thank you for your careful review and positive assessment of our work! We are particularly encouraged by the recognition of the paper's central probabilistic perspective and the empirical support for this view across multiple tasks and metrics. We address your comments and questions below.
>
> **Weaknesses:**
>
> > Lack of theory foundation while being empirically-focused
>
> We agree that a formal theoretical framework is an important direction for future work, and believe our empirical findings point toward promising directions for developing one. Our current view is that bimodality reflects competition between distinct solution types near a task's minimum capacity threshold. Random seeds do not just add variance to the training outcome but they can alter which loss basin training enters among competing circuits. Model scale impacts emergence in two ways: first, it can cross a minimum-capacity boundary where a successful circuit becomes representable (e.g., induction heads require at least two attention layers, making single-layer models categorically incapable regardless of seed); second, beyond that boundary, larger models increase the probability that optimization finds and stabilizes that circuit. Formalizing this via connections to the lottery ticket hypothesis or grokking dynamics is an exciting direction we hope our empirical characterization motivates. We will include a dedicated limitations section in the camera-ready version addressing this gap.
>
> > The empirical analysis focuses primarily on single-level bimodal splits within individual tasks. The paper does not investigate whether emergence may involve multi-stage or hierarchical bifurcations across interacting skills. As a result, the generality of the proposed bimodality framework … remains unclear.
>
> For tasks requiring multiple hierarchical skills, models from different seeds might acquire different subsets, producing more complex multimodal distributions. Our synthetic algorithmic tasks provide a clean way to study this: we can train on mixtures (e.g., counting, addition, and a combined task requiring both), then study whether each skill exhibits independent bimodal variation and whether the joint distribution becomes multimodal. We thank the reviewer for pointing out this natural extension of our framework.
>
> **Questions:**
>
> > What conceptual and methodological challenges would arise in extending the proposed bimodality framework from single-level splits to multi-stage or hierarchical bifurcations of capabilities?
>
> When multiple skills compose, the joint distribution may no longer yield a clean bimodal split. For complex tasks like GSM8K, disentangling which skill drives each bifurcation would be difficult. In synthetic settings we proposed above, this becomes tractable: separate test sets for each skill (e.g., counting, addition) and their combination can diagnose whether multimodality arises independently per skill or only in composition.
>
> > Is unimodality at small scales a theoretical necessity of the proposed framework, or merely an empirical observation within the tested seed budget?
>
> We believe it is a theoretical necessity. If a model lacks the minimum capacity to represent any successful solution, no seed can find one, and all runs fail uniformly — producing a unimodal distribution at chance. Bimodality can only emerge once the minimum capacity threshold is crossed and seeds begin to diverge in whether they discover and stabilize a successful circuit.
>
> **Limitations:**
>
> > The paper does not include a dedicated Limitations section. While certain constraints are implicitly acknowledged…they are not systematically discussed.
>
> We will use the extra camera-ready page to thoroughly address limitations. Below is the outline:
>
> **Computational and experimental constraints**. Our LM experiments rely on partially reinitialized Qwen2.5 models rather than full pretraining from scratch, introducing randomness that could amplify or suppress bimodality. Testing across other families (e.g., Llama, Phi, Mistral) would strengthen generality.
>
> **Task coverage**. Our MMLU continued pretraining mostly recovers multiple-choice formatting; full capabilities are difficult to recover given our pretraining set (~ millions of tokens) versus Qwen's original corpus (~ billions). This also explains why we do not examine tasks like GSM8K: achieving non-trivial reasoning across 80 seeds would be prohibitively expensive.
>
> **Sensitivity to training choices**. We do not address whether bimodality is sensitive to hyperparameter choices such as learning rate, batch size, or optimizer. If bimodality proves fragile, the practical implications would be narrower than suggested.
> Theory and mechanistic understanding. We document when and where bimodal distributions appear but do not provide a formal theoretical framework for when or why bimodality must arise. Developing formal conditions for this is an important direction for future work.

---

> > ### Author Rebuttal · Reviewer_CftN · 2026-04-04
> >
> > The rebuttal resolved my concerns and I'll raise my score.

---

### Official Review · Reviewer_5uG2 · 2026-02-26

**Soundness:** 4
**Presentation:** 4
**Significance:** 4
**Originality:** 3
**Overall Recommendation:** 5
**Confidence:** 3

**Summary:**

This paper proposes a novel distributional perspective to explain the "emergence" phenomena in large models.
The authors clarify that the 'breakthrough' is an illusion created by sampling from two distinct performance clusters. As the distribution is bimodal, the observed jump occurs when one sample is drawn from the 'failure' peak and a subsequent sample is drawn from the 'success' peak.
The paper provides extensive empirical evidence across algorithmic tasks and natural language benchmarks to support this claim. Furthermore, the authors investigate the underlying "competition" between different rules/features to explain why such bimodal variance exists.

**Compliance With Llm Reviewing Policy:**

Affirmed.

**Final Justification:**

My final recommendation is score 5 (Accept).

This paper presents a highly **original** and **significant** distributional perspective on the "emergence" phenomenon in large models. By focusing on performance variance and bimodality, the authors provide a more nuanced understanding of scaling laws than traditional deterministic views. The work is exceptionally **sound**, with well-supported conclusions drawn from a diverse set of experiments. Furthermore, the **clarity** of the writing and the logical structure of the arguments make the complex topic of mechanistic interpretability highly accessible.

The authors' rebuttal **fully addressed my concerns** regarding the interpretative nature of the "bottleneck loss" metric and the seemingly contradictory effects of scaling in different task settings (English syntax vs. MMLU).

**Key Questions For Authors:**

1. In Figure 7, the authors explain the bimodality of the 6-layer model by suggesting that larger models tend to "grok" one specific rule while ignoring others (i.e., rule competition). However, in Figure 5(b), increasing the model scale seems to lead to the consistent and "perfect" acquisition of MMLU capabilities, shifting away from bimodality toward a successful unimodal distribution. **Could the authors provide an explanation for these seemingly opposite effects of scaling?** Why does increased scale exacerbate rule competition/bimodality in English syntax acquisition tasks but resolve it in the MMLU setting?

**Limitations:**

The authors could further discuss the discrepancy in how scaling affects bimodality across different task categories, as detailed in the Questions section.

**Strengths And Weaknesses:**

**Strengths**

1. The authors move beyond the traditional "capability" and "metric-mirage" debates by focusing on performance variance. This distributional view provides a more nuanced understanding of how scaling affects model behavior stochastically rather than deterministically.
2. A significant contribution of this work is the insight that smaller models may already possess the latent "capability" to generalize, but they simply fail to do so consistently due to high variance. This opens up a promising research direction on how to reliably elicit these capabilities in smaller-scale models.
3. The work is highly complete. It identifies the "jumpy" performance (explained by variance) and attempts to explain the source of this variance through the lens of feature competition. The conclusions are well-supported by a diverse set of experiments.
4. The paper is well-structured, the arguments are logically sound, and the writing is easy to follow, making it accessible to both researchers studying scaling laws and those focused on mechanistic interpretability.

**Weaknesses**

1.  In Section 2.4, the authors introduce a metric defined as the mean of the maximum per-token loss in Eq~(1). While the authors discuss the calculation and the continuity of this metric, there is a lack of deep discussion regarding its interpretative nature. It is unclear what specific property of model behavior this "bottleneck" loss ($\max L$) captures that standard metrics (e.g., NLL Loss) do not, and whether this specific choice of metric might inadvertently amplify the bimodal characteristics observed in the results.

Typo:
1. In the caption of Figure 5, "(c) For a fixed data mix," appears to be corrected to "(b)".
2. There is a repetition on page 6, line 323. The sentence *"In Figure 8, negative log likelihood (NLL) loss remains bimodal"* appears twice consecutively.

---

> ### Author Rebuttal · Authors · 2026-03-30
>
> Thank you for your thoughtful feedback and positive assessment of our work! We are grateful for your recognition of the importance in studying variance in understanding emergence in scaling and the completeness of the empirical analysis. We address your comments and questions below.
>
> **Weaknesses:**
>
> > In Section 2.4, the authors introduce a metric defined as the mean of the maximum per-token loss in Eq~(1). While the authors discuss the calculation and the continuity of this metric, there is a lack of deep discussion regarding its interpretative nature. It is unclear what specific property of model behavior this "bottleneck" loss () captures that standard metrics (e.g., NLL Loss) do not, and whether this specific choice of metric might inadvertently amplify the bimodal characteristics observed in the results.
>
> Thank you for raising this question, it is an important clarification. The maximum per-token loss is meant to capture whether the model has certain failure points in the sequence, which standard averaged metrics such as NLL can obscure. On these tasks, most tokens are often easy, so averaging over all positions smooths away the key distinction between runs that generalize and runs that fail at a specific step. This metric removes much of this background noise and makes the underlying structure easier to detect; in that sense, we do not view it as creating bimodality, but rather as reducing the number of model samples needed to observe it clearly.
>
> Thank you for catching those typos, we will be sure to correct them!
>
> **Questions:**
>
> > In Figure 7, the authors explain the bimodality of the 6-layer model by suggesting that larger models tend to "grok" one specific rule while ignoring others (i.e., rule competition). However, in Figure 5(b), increasing the model scale seems to lead to the consistent and "perfect" acquisition of MMLU capabilities, shifting away from bimodality toward a successful unimodal distribution. Could the authors provide an explanation for these seemingly opposite effects of scaling? Why does increased scale exacerbate rule competition/bimodality in English syntax acquisition tasks but resolve it in the MMLU setting?
>
> This is a great question and the answer lies in the different nature of the two settings.
>
> In the MMLU setting, the training data mixes C4 news with MMLU examples, and the model must balance between general language modeling (i.e., C4) with MMLU-specific capabilities (multiple choice format and language understanding). Such a setting implies that when a model fails on MMLU, it simply fails to acquire the capability. Therefore, bimodality reflects a split between models that recovered MMLU ability and those that did not, and larger models resolve this bimodality by acquiring the capability more reliably. This contrasts with the case below where, when models acquire a certain capability, there are two competing solutions for this capability, leading to different test (OOD) performances.
>
> In the English grammar acquisition task, the training data is intentionally ambiguous between two mutually exclusive grammar rules: the hierarchical rule and the linear rule. Learning either rule can achieve perfect in-distribution accuracy but produce opposite out-of-distribution predictions, creating a rule competition. Small models lack the capacity to commit to either rule, producing unimodal intermediate performance. Larger models can fully commit to one rule, but which rule they choose still varies across seeds, leading to bimodality in Figure 7.
>
> In summary, in the MMLU case, model scale resolves bimodality when there is no competing alternative solution. In contrast, model scale sharpens bimodality when the task involves mutually exclusive strategies (syntax acquisition). We thank the reviewer for raising this question. This question highlights an important distinction between two mechanisms underlying the rise of bimodality, and we will discuss it in the camera-ready version.
>
> **Limitations**
>
> See above.
>
> Several reviewers have pointed out that this paper would benefit from a more detailed discussion of limitations. We plan to use the additional page allowed in the camera-ready version to thoroughly address these. Specifically, we will discuss: (1) the computational and experimental constraints of our setup, including the use of partial reinitialization rather than full pretraining and the restriction to a single model family; (2) the limited task coverage, particularly why recovering deeper capabilities like mathematical reasoning on GSM8K is challenging at the seed density our study requires; and (3) the absence of a formal theoretical framework for predicting when and why bimodality arises, as well as the sensitivity of our findings to hyperparameter choices beyond architecture.

---

> > ### Author Rebuttal · Reviewer_5uG2 · 2026-04-03
> >
> > I would like to thank the authors for their detailed response and the additional analysis regarding the two mechanisms underlying the rise of bimodality. The clarification has addressed my initial concerns. I maintain my current score 5.

---

### Official Review · Reviewer_ELzw · 2026-03-13

**Soundness:** 4
**Presentation:** 4
**Significance:** 3
**Originality:** 3
**Overall Recommendation:** 5
**Confidence:** 3

**Summary:**

This work finds a reconciliation between the “emergent abilities are real” and “emergent abilities are a mirage” camps by discovering that apparent emergent abilities arise from bimodal performance distributions across training seeds. As model scale increases, the probability that a training run acquires a capability increases gradually. When only a single seed is evaluated per model size, the sampled run may fall into either the high-skill or low-skill mode, producing either smooth or discontinuous scaling curves. The authors demonstrate this phenomenon on the Qwen model family for both synthetic length generalization tasks, as well as MMLU and English syntax acquisition.

**Compliance With Llm Reviewing Policy:**

Affirmed.

**Final Justification:**

The authors included more detailed discussion of limitations. My concerns have been addressed, and I leave my score at 5.

**Key Questions For Authors:**

The causal mechanism behind the bimodal skill distributions remains somewhat unclear. Have the authors thought about mechanistic analysis of training dynamics and model behavior to pinpoint this phenomenon more precisely?

**Limitations:**

See weaknesses section.

**Strengths And Weaknesses:**

# Strengths

This paper is very well written and covers a lot of ground in just eight pages. The work also provides a compelling reconciliation of two competing explanations for emergent abilities, which can help alleviate debate that might otherwise stymie progress in this area. Experiments are done both on fully controllable synthetic tasks (counting, addition) and real-world tasks like MMLU and English syntax acquisition. Lastly, the authors highlight an important methodological issue in scaling studies: evaluating only a single training seed per model size can produce misleading conclusions about capability emergence. The findings and issues raised here are significant.


# Weaknesses:
- Figure 1 could benefit from clearer labeling of modes - it takes a little reading to understand exactly what is meant by “bimodal” here. Reader comprehension could be significantly enhanced if the modes in the histograms had large text specifying “mode 1” and “mode 2,” immediately commanding visual attention to the phenomenon.
- Experiments are only done on the Qwen model family. Since model families vary in their training data, architecture choices, etc., for this work specifically, it is probably more important to test *across* open-source model families (e.g. Llama, Phi, Mistral, etc.) rather than within.
- Many emergence claims involve reasoning or compositional tasks (e.g., GSM8K) that are not examined here. While the paper already covers a wide range of experiments, a more explicit discussion of the scope and limits of the findings would strengthen the paper.

---

> ### Author Rebuttal · Authors · 2026-03-30
>
> Thank you for your thoughtful feedback and positive assessment of our work; we're particularly encouraged by the recognition of the paper's central contribution, breadth of experiments, and methodological significance! We address your comments and questions below.
>
> **Weaknesses:**
>
> > Figure 1 could benefit from clearer labeling of modes
>
> Thank you for your detailed feedback! We agree that this will make the phenomenon more clear and will update the figure according to your suggestion!
>
> > Experiments are only done on the Qwen model family. Since model families vary in their training data, architecture choices, etc., for this work specifically, it is probably more important to test across open-source model families (e.g. Llama, Phi, Mistral, etc.) rather than within.
>
> Testing across model families would indeed strengthen our claims. Qwen models were chosen because they are the only family with small-scale models (<1B) achieving non-trivial performance on "emergent" benchmarks like MMLU — a prerequisite for observing bimodality in our continued pretraining setup. Other small-scale models (e.g., Llama) have near-trivial base MMLU performance, making our reinitialization-and-recovery design less informative since there is little capability to recover.
>
> That said, preliminary results from concurrent work [1] suggest bimodal performance distributions also arise in Llama 8B under LoRA fine-tuning across data scales, consistent with our findings. We plan to include Llama experiments based on [1] in the camera-ready appendix and have discussed this limitation explicitly in our proposed limitations section (see below).
>
> > Many emergence claims involve reasoning or compositional tasks (e.g., GSM8K) that are not examined here. While the paper already covers a wide range of experiments, a more explicit discussion of the scope and limits of the findings would strengthen the paper.
>
> MMLU is a canonical task exhibiting emergence [2], motivating our task choice. We will use the camera-ready extra page to thoroughly address the paper's limitations. The "Task Coverage" section discusses why reasoning tasks like GSM8K are excluded. Below is an outline of the section:
>
> __Computational and experimental constraints__. Ideally, we would pretrain full models from scratch across many scales and seeds, but this is prohibitively expensive. Our LM experiments instead rely on partially reinitialized Qwen2.5 models, introducing randomness that differs from training from scratch and could amplify or suppress bimodality. Testing across other families (e.g., Llama, Phi, Mistral) would strengthen generality.
>
> __Task coverage__. Our MMLU continued pretraining mostly recovers multiple-choice formatting; full language modeling capabilities are difficult to recover given our pretraining set (~ millions of tokens) versus Qwen's original corpus (~ billions). This resource gap also explains why we do not examine tasks like GSM8K: achieving non-trivial mathematical reasoning across 80 seeds would be prohibitively expensive.
>
> __Sensitivity to training choices__. We do not address whether bimodality is sensitive to hyperparameter choices such as learning rate, batch size, or optimizer. If bimodality proves fragile to these choices, the practical implications would be narrower than suggested.
>
> Theory and mechanistic understanding. We document when and where bimodal distributions appear but do not provide a formal theoretical framework for when or why bimodality must arise, nor fully characterize its mechanistic origin. Developing formal conditions for this is an important direction for future work.
>
> **Questions:**
>
> > Have the authors thought about mechanistic analysis of training dynamics and model behavior to pinpoint this phenomenon more precisely?
>
> This is an exciting direction. Our current view is that bimodality reflects competition between distinct solution types near a task's minimum capacity threshold — random seeds can alter which basin of attraction training enters among competing circuits, not merely add variance. Scale helps by crossing the minimum-capacity boundary where a successful circuit becomes representable, and by increasing the probability that optimization finds and stabilizes it.
>
> Promising mechanistic analyses include: tracking training trajectories around the onset of bimodality to identify when runs diverge; localizing heads/circuits whose behavior separates successful from unsuccessful runs (e.g., testing whether patching or reinitializing them can switch a run's mode); and comparing circuit formation across competing solution types in the paper.
>
> [1] Phase Transitions in Backdoor Learning: Minimum Data Poisoning Thresholds for LLM Backdoors (https://boazbk.github.io/mltheoryseminar/student_projects/final_papers_and_posters/papers/Phase_Transitions_in_Backdoor_Learning__Minimum_Data_Poisoning_Thresholds_for_LLM_Backdoors_(2)_-_Kaden_Zheng.pdf)
>
>
> [2] Emergent Abilities of Large Language Models (https://arxiv.org/pdf/2206.07682)

---

> > ### Author Rebuttal · Reviewer_ELzw · 2026-03-31
> >
> > Thank you for the response and more detailed discussion of limitations. My concerns have been addressed, and I leave my score at 5.

---

### Decision · Program_Chairs · 2026-04-30

**Decision:**

Accept (regular)

**Comment:**

This paper reignites the "emergence is a mirage" discussion and finds a reasonable middle ground which seems to mechanistically explain this phenomenon as being due to bimodality of a performance metric between two random execution environments (RNG seeds). Insights and claims are supported by a wide array of experiences.

After a long and rich discussion between the authors and reviewers, there is concensus among the reviewers to accept this paper to ICLR. Therefore, I'm recommending acceptance.